# Novel Supercapacitor Electrode Derived from One Dimensional Cerium Hydrogen Phosphate (1D-Ce(HPO_4_)_2_.xH_2_O)

**DOI:** 10.3390/molecules27227691

**Published:** 2022-11-09

**Authors:** Jong Hee Yoon, Bak Jinsoo, Inho Cho, Rajangam Vinodh, Bruno G. Pollet, Rajendran Suresh Babu, Hee-Je Kim, Sungshin Kim

**Affiliations:** 1Department of Electrical and Computer Engineering, Pusan National University, Busan 46241, Korea; 2Green Hydrogen Lab (GH2Lab), Institute for Hydrogen Research (IHR), Université du Québec à Trois-Rivières (UQTR), 3351 Boulevard des Forges, Trois-Rivières, QC G9A 5H7, Canada; 3Laboratory of Experimental and Applied Physics, Centro Federal de Educação Tecnológica Celso Suckow da Fonseca, Av. Maracanã Campus 229, Rio de Janeiro 20271-110, Brazil

**Keywords:** cerium hydrogen phosphate, nanorods, specific energy, specific power, stability

## Abstract

In this manuscript, we are reporting for the first time one dimensional (1D) cerium hydrogen phosphate (Ce(HPO_4_)_2_.xH_2_O) electrode material for supercapacitor application. In short, a simple hydrothermal technique was employed to prepare Ce(HPO_4_)_2_.xH_2_O. The maximum surface area of 82 m^2^ g^−1^ was obtained from nitrogen sorption isotherm. SEM images revealed Ce(HPO_4_)_2_.xH_2_O exhibited a nanorod-like structure along with particles and clusters. The maximum specific capacitance of 114 F g^−1^ was achieved at 0.2 A g^−1^ current density for Ce(HPO_4_)/NF electrode material in a three-electrode configuration. Furthermore, the fabricated symmetric supercapacitor (SSC) based on Ce(HPO_4_)_2_.xH_2_O//Ce(HPO_4_)_2_.xH_2_O demonstrates reasonable specific energy (2.08 Wh kg^−1^), moderate specific power (499.88 W kg^−1^), and outstanding cyclic durability (retains 92.7% of its initial specific capacitance after 5000 GCD cycles).

## 1. Introduction

Supercapacitors (SCs), also called electrochemical capacitors, are largely applied in different power-saving sectors such as continuous power transfers and transportable electronic equipment [1,2,3,4,5]. The distinct benefits of SCs are quick charge/discharge capacity, excellent specific power, and remarkable capacitance retention (stability) [6]. Despite all these merits, SCs hurt by poor specific energy that causes difficulties in development [7]. Generally, electrode materials play a crucial part in SCs as charge storage at the interface of electrode and electrolyte. Hence, the well-organized structure and distinct surface morphology of the electrode materials have a prodigious influence on the capacitance and enactment of the electrodes [8,9]. For instance, the pore size distribution in SC electrodes involved a prominent role in the loading, mobility, and transport of ions, and these factors define the rate, efficiency, and enactment of the supercapacitor electrode materials [10]. Thus, it is important to attain the unique architecture and configuration for electroactive materials leading to an SC electrode with outstanding behavior.

Recently, cerium (Ce)-based electrode materials have been studied extensively for supercapacitor applications. Ce possesses excellent specific capacitance properties due to its extraordinary re-dox capability between Ce^3+^ and Ce^4+^. The unique size and surface morphology of various cerium-derived electrode materials played crucial parts in the supercapacitor performances. Although cerium is a very active rare earth metal, the rates of re-dox electrochemical reactions taking place in the bulk materials were sluggish, which hinders its practicality for electrochemical energy storage equipment. In order to obtain high-performance supercapacitors, many researchers focused their minds on synthesizing special morphology of Ce-derived electrode materials. In recent days, CeO_2_ (cerium oxide) is an efficient constituent for the construction of the electrodes of supercapacitors. For example, Abdul et al. synthesized hierarchical pores of cerium oxide nanoparticles by facile precipitation followed by a hydrothermal technique, which displays an extreme specific capacitance of 877.5 F g^−1^ at 3 A g^−1^ as well as outstanding rate capability [11]. Bhusankar et al. synthesized CeO2 sheets with a rhombus shape by a simple methodology [12]. The prepared supercapacitor material showed the highest capacitance of 481 F g^−1^ at 5 mV s^−1^. Furthermore, it retains approximately 83% of its original capacitance over five hundred cycles. Nevertheless, the poor electrical conductivity of cerium oxide limits its electrochemical performance, leading to inferior cyclic performance and minimal specific capacitance. The poor electrical conductivity of CeO_2_ is generally rectified by making composites with conductive materials and organic binders on the substrate [13]. For instance, Hu et al. synthesized Ce-derived CeMO_3_ (M = Ni, Cu, Co) perovskites by electrospinning method. The maximum specific capacitances of CeNiO_3_, CeCuO_3_, and CeCoO_3_ perovskites were attained for 189, 117, and 128 F g^−1^ at 0.5 A g^−1^, respectively. The authors concluded these SC electrode materials were efficient and could be applied for pilot-scale supercapacitor applications in near future [14].

Very recently, Usman et al. prepared a nanocomposite of CeO_2_/rGO/CeS_2_ electrode for supercapacitor applications [15]. The as-prepared pores enriched composite of CeO_2_/rGO/CeS_2_ established the supreme capacitance of 720 Fg^−1^, with the highest specific energy (23.5 Wh kg^−1^), and specific power (2917.2 W kg^−1^), respectively, at 2.5 Ag^−1^. In addition, the prepared composite electrode exhibited an outstanding cycle life after 3000 cycles at 100 mV s^−1^. Wang et al. reported nanocomposite of CeO_2_/graphene for supercapacitor application was studied by depositing cerium oxide nanoparticles onto three-dimensional graphene material [16]. Further, the composite CeO_2_/graphene shows a maximum capacitance (208 F g^−1^) and specific power (~18 kW kg^−1^), illustrating a robust synergistic effect probably subsidized by the enhanced electrical conductivity of cerium oxide and improved consumption of graphene.

Aravinda et al. fabricated CeO_2_/AC (activated carbon) derived composite electrode by a facile mechanical mixing procedure [17]. The composite of 10 wt% CeO_2_ shows a specific capacitance of 162 F g^−1^ in a two-electrode arrangement. The assembled configuration holds 86% of its initial capacitance and remains at high current density with outstanding cyclic stability. Furthermore, the fabricated composite displays a specific power of 3500 W kg^−1^ at a higher current density, illustrating an efficient electrode material for SC applications. Instantaneously, the nano CeO_2_/AC composite displayed a specific capacitance of 162 F g^−1^. Padmanathan et al. fabricated NiO–CeO_2_ binary oxide for ultracapacitor via. glycol-mediated citrate sol–gel technique. Further, the prepared NiO–CeO_2_ pyrolyzed at 500 °C exhibits a maximum capacitance of 305 F g^−1^ at 1 A g^−1^ [18].

As far as we can know, there is almost no paper reported about the application of cerium hydrogen phosphate (Ce(HPO_4_)_2_.xH_2_O) with a specific structure as a supercapacitor electroactive material. Recently the size and shape-dependent behaviors of cerium-based materials have fascinated more consideration due to their excellent scientific applications. The present work focused on the synthesis of one-dimensional Ce(HPO_4_)_2_.xH_2_O via a hydrothermal method and employing it as a super capacitance electrode. The morphology and surface area of the synthesized material was studied by different physicochemical characterizations, as were their specific capacitance characteristics by electrochemical methods. The supercapacitor outcomes established that the cerium hydrogen phosphate electrode material showed adequate supercapacitor performance. Furthermore, the constructed SSC demonstrated the highest specific capacitance of 23.44 F g^−1^ at 0.3 A g^−1^.

## 2. Experimental Section

### 2.1. Materials

Ninety-nine percent of Ce(NO_3_)_2_.6H_2_O and ninety-nine percent of Na_2_HPO_4_ were acquired from Samchun chemicals, Korea. The substrate material, nickel foam (NF) for the fabrication of working electrode (WE) was purchased from Sigma Aldrich, St. Louis, MO, USA.

### 2.2. Synthesis of Cerium Hydrogen Phosphate

Cerium hydrogen phosphate was prepared by a facile hydrothermal technique. In brief, the required quantity of cerium nitrate (2.6 mmol) and deionized water (15 mL) were taken in a 100 mL glass beaker and allowed to stir for 30 min at room temperature. Then, the completely soluble disodium hydrogen phosphate (2.6 mmol) in 15 mL DI water was slowly incorporated into the aforementioned reaction content under continuous stirring. After complete dissolution, the whole reaction content was moved into the stainless-steel (SS) autoclave and carried out the reaction at 120 °C for 12 h. The SS autoclave was permitted to cool at ambient temperature once the reaction was completed. Then, the resultant content was washed with an abundant amount of ethanol and water in a filtration unit coupled with a vacuum pump. Finally, the Ce(HPO_4_)_2_.xH_2_O was desiccated at 100 °C for 18 h. The comprehensive reaction pathway was depicted in Figure 1.

### 2.3. Physiochemical Characterization

Fourier transform infra-red was used to envisage the creation of novel functional groups in the produced Ce(HPO_4_)_2_.xH_2_O (FT-IR; Perkin Elmer, Waltham, MA, USA). The morphology of the prepared Ce(HPO_4_)_2_.xH_2_O was characterized by scanning electron microscopy (SEM; Quattro S, Singapore) analysis. The textural property of the synthesized sample was determined by nitrogen adsorption/desorption isotherm of Brunauer–Emmet–Teller method (BET: BELSorp Max, Osaka, Japan). X-ray diffraction pattern was studied to explore the structure of crystalline materials (XRD; SmartLab, Tokyo, Japan) array. The oxidation state and existence of the elements in the Ce(HPO_4_)_2_.xH_2_O was evaluated by X-ray photoelectron spectroscopy (XPS; Thermo Fisher Scientific, Oxford, UK). The thermal stability of the Ce(HPO_4_)_2_.xH_2_O materials was analyzed by thermogravimetric analysis (TGA; Mettler-Toledo, Greifensee, Switzerland).

### 2.4. Electrode Fabrication

Prior to the electrode fabrication, NF was dressed with 3% dil. HCl, acetone, and DI water for 10 min each in an ultra-sonication tub and then dried at 80 °C for 12 h. The working electrode was constructed by mixing the Ce(HPO_4_)_2_.xH_2_O nanorods, PVDF, and acetylene black with a ratio of 85:5:10 with few drops of N-methyl pyrrolidine included gaining a homogeneous mixture. Then, it was coated on a pre-cleaned nickel foam with an active area of 1 × 1 cm^2^. The constructed WE was desiccated in an oven at 80 °C for 12 h. The weight of the active material presented in the nickel foam was ~4 mg.

The electrochemical properties of the constructed WE were evaluated by cyclic voltammetry (CV), galvanostatic charge-discharge (GCD), and electrochemical impedance spectroscopy (EIS) methods using Biologic-150 electrochemical workstation. The organized three-electrode configuration contains Ce(HPO_4_), Pt wire, and Hg/HgO as the working electrode (WE), counter electrode (CE), and reference electrode (RE), respectively. The entire electrochemical reaction was executed in a 3 M KOH supporting electrolyte. EIS was achieved in the frequency range from 0.01 Hz to 0.1 kHz.

The *C*_s_ of the constructed working electrode, Ce(HPO_4_)_2_.xH_2_O /NF was assessed from the discharge time of the GCD curve [19,20] using the subsequent Formula (1).
(1)$$C_{s}=\frac{I\int vdt}{w\times \Delta V}$$
where *I* represents applied current in amps, ∫ vdt represents the area under the discharge curve in seconds, *w* signifies the mass of the Ce(HPO_4_)_2_.xH_2_O in grams, and Δ*V* indicates the potential range in volts.

The specific capacitance of the constructed SSC [21] was evaluated by the upcoming Equation (2),
(2)$$\;C_{cell}=\frac{I\times \Delta t\;}{m\times \Delta V}$$
where *I* represents applied current (A), Δt implies the discharge time from the GCD curve (s), *m* indicates the mass of the two electrodes summed (g), and Δ*V* indicates the potential range in volts.

Moreover, the specific energy (Wh kg^−1^) and specific power (W kg^−1^) are crucial parameters to compute in the analysis of the electrochemical evaluation of Ce(HPO_4_)_2_.xH_2_O)//Ce(HPO_4_)_2_.xH_2_O SSC device [22]. The *Ragone* plot of Ce(HPO_4_)_2_.xH_2_O//Ce(HPO_4_)_2_.xH_2_O SSC device is attained positioned on the GCD results by means of the following Equations (3) and (4).
(3)$$E_{d}=\frac{C_{cell}\times \Delta V^{2}\;}{7.2}$$
(4)$$P_{d}=\frac{3600\times E\;}{\Delta t}$$

## 3. Results and Discussion

The surface morphology of the as-synthesized Ce(HPO_4_)_2_.xH_2_O was analyzed using SEM as exposed in Figure 1a–c with diverse magnifications. A rod-like Ce(HPO_4_)_2_.xH_2_O nanorod has grown on the Ni foam surface with a tiny amount of particles and clusters. Furthermore, it was noticed that the sample surface is smooth with an average diameter of ~75 to 150 nm and a length of ~400 to 600 nm. Furthermore, the composition of the prepared Ce(HPO_4_)_2_.xH_2_O was analyzed by the energy-dispersive X-ray analyzer (EDAX), to recognize the constituent of the Ce, P, and O in the prepared material as shown in Figure 1d. There was no impurity notified in the EDX and these quantitative data confirm the constituent, purity, and formation of Ce(HPO_4_)_2_.xH_2_O.

In order to confirm the existence of a new functional moiety, the as-prepared sample was examined by FT-IR (Figure 2a) in the region between 4000 and 400 cm^−1^. The FT–IR spectrum displayed broad peaks of approximately 435–690 cm^−1^ owing to the PO_4_ stretching vibration. The other peaks ascribed to the O-H stretching and bending vibrations of water adsorbed on the Ce(HPO_4_)_2_.xH_2_O surface were also identified at 1612 and 3494 cm^−1^, correspondingly [23]. It is worth stating that the existence of adsorbed water might assist to enhance the electrochemical performance by increasing interparticle distance, which disparately adjusts the ion transport pathways [24,25]. XRD pattern was studied to examine the crystalline property of the synthesized Ce(HPO_4_)_2_.xH_2_O and it was demonstrated in Figure 2b. The peaks at 2θ values of 25°, 28°, 33°, 37.5°, 42°, 46°, 52°, and 54° correspond to (110), (200), (102), (112), (211), (212), (220), and (310) planes, respectively. It clearly indicates the peaks are well-matched with the cerium phosphate hydrate [26,27]. Moreover, the hydrothermal treatment supported the crystalline nature of cerium hydrogen phosphate. The surface properties of the prepared Ce(HPO_4_)_2_.xH_2_O material were evaluated by the nitrogen adsorption/desorption isotherms [28]. The nitrogen sorption isotherm of Ce(HPO_4_)_2_.xH_2_O is depicted in Figure 2c. A steady rise in the amount of adsorbed N_2_ can be seen when the relative pressure (P/P_0_) increases from 0.0 to 1.0. As per the IUPAC classification, the shape of the sample is attributed to type IV isotherm. A small rise at low relative pressure signifies the presence of micropores, while the sharp increase from P/P_0_ = 0.45 to 1.0 indicates the abundance of mesopores [29]. The mesoporous structure reduces the diffusion distance of electrolyte ions [30]. The mesopores have diameters between 10 and 50 nm, and they can provide a short ion-transport pathway through the walls, with a minimized inner-pore resistance, which is beneficial for supercapacitor applications [31,32]. The BET-specific surface area of the Ce(HPO_4_)_2_.xH_2_O is calculated as 85 m^2^ g^−1^. The thermal characteristics of the prepared material were analyzed by TGA and it was illustrated in Figure 2d. The first weight residue was perceived at ~100 to 130 °C attributed to the vaporization of coordinated water. The second weight residue at ∼490 °C was ascribed to the thermal cleavage of the main phosphate network. Moreover, the supreme weight loss T_max_ of ~14.67 % was attained at 350 °C and the end ∼6.9 % of mass residue was acquired at 800 °C. The well-aligned peak noticed at 540 °C of DTGA signifies that the Ce(HPO_4_)_2_.xH_2_O material possesses crystalline behavior. The outcomes reveal that the prepared Ce(HPO_4_)_2_.xH_2_O are extremely stable and can be usable in high-temperature applications [33].

Figure 3a shows the survey spectrum of Ce(HPO_4_)_2_.xH_2_O, which reveals the elements of Ce, P, and O. Figure 3b–d represents the high-resolution XPS spectra of O *1s*, P *2p*, and Ce *3d*, respectively. The high-resolution XPS spectrum of O 1*s* is deconvoluted into two distinct peaks at 531.6 and 533.2 eV (Figure 3b), which can be associated with the phosphate and coordinated water in the synthesized Ce(HPO_4_)_2_.xH_2_O [34]. The P 2*p* high-resolution XPS spectrum of the Ce(HPO_4_)_2_.xH_2_O (Figure 3c) shows two deconvoluted peaks at 132.9 and 133.8 eV accountable for the P 2*p*_3/2_ and P 2*p*_1/2_ of elemental P state and oxidized P state, respectively [35]. In Figure 3d, six binding energy peaks ofthe typical Ce *3d* occurs at 881.9, 888.6, 897.1, 901.4, 907.9, and 916.2 eV in that the first three peaks are associated with Ce*3d*_5/2_ and the second three peaks are associated with Ce *3d*_3/2_. The peak positions of Ce *3d*_5/2_ and Ce *3d*_3/2_ corresponded to the spin–orbit features of Ce^4+^ [36].

The electrochemical behavior of the Ce(HPO_4_)_2_.xH_2_O/NF electrode was initially measured by three-electrode cell arrangement using a 3 M KOH solution. The counter electrode (CE) and reference electrode (RE) were platinum wire and Hg/HgO, correspondingly. Figure 4a illustrates the typical voltammogram of the Ce(HPO_4_)_2_.xH_2_O/NF electrode with various sweep rates extending from 2 to 300 mV s^−1^ in the cut-off potential margin between −0.4 and 0 V vs. Hg/HgO. The shape of the CV plots with a pair of oxidation and reduction peaks evidently reveals the pseudo-capacitance properties. The pair of peaks is primarily related to the Faradaic redox reaction corresponding to Ce^3+^ to Ce^4+^ [37,38,39]. The anodic and cathodic current densities increase when the sweep rate increases. It can be seen that the profile of the CV is well-maintained the redox peaks increasing the sweep rates, which resulting a rapid CV response towards a quick scan of the potential window and an outstanding rate capability for energy storing. Figure 4b presents the GCD plots of Ce(HPO_4_)/NF electrode at different densities of current. During the discharging process, the ohmic drop was obtained, which is normally caused by the electrode’s internal resistance, which is quite low around 50 mV at 0.2 A g^−1^. This suggests exceptional interfacial interaction between the active component and the NF along with lower over potential during the GCD process. To study the crucial performance of the Ce(HPO_4_)_2_.xH_2_O/NF electrode for supercapacitor, EIS measurements were studied by the Nyquist graph, which demonstrates the frequency response of the electrolyte/electrode process and is commonly schemed against the real part (*Z*′) versus imaginary part (*Z″*) of the impedance. Figure 4c presents the fitted Nyquist plot of the Ce(HPO_4_)/NF electrode, which is usually inferred by fitting the experimental data by an equivalent electrical circuit. The initial intercept of the semicircle with the *Z*′ and the value of both internal resistance electrode material and ohmic resistance of the electrolyte are corresponding to solution resistance (*R*_s_). In the higher frequency range, the semicircle might be an interfacial charge-transfer resistance (*R*_ct_). The mid-frequency region shows the straight line having a slope of 45°, which signifies the Warburg resistance (*W*_d_) and the pseudo-capacitance *C*_P._ The inset of Figure 4c shows the suitable equivalent circuit for Ce(HPO_4_)_2_.xH_2_O/NF electrode. The lowest *R*_s_ value was obtained for Ce(HPO_4_)_2_.xH_2_O/NF electrode and denotes the good conductivity and higher specific capacitance. The electrode achieved very low *R*_ct_ and *W*_d_ values. The low value for *W*_d_ denotes a small diffusion path of the ions in electrolytes within the composites. The major idea for a supercapacitor is the factor of frequency (*n*), which can give details on the ideality of material towards supercapacitive behavior. The values of *n* vary between 0 and 1: *n* = 0 designates the resistor; *n* = 1 designates the ideal capacitor. The Ce(HPO_4_)_2_.xH_2_O/NF electrode demonstrated reasonable behavior of the supercapacitor and attained the high *n* value of 0.70.

In order to explore the mechanism of the charge storage of the prepared electrode, we employed power law (*i* = *aνb*) [40] to evaluate the connection between the obtained current (*i*) and sweep rate (*ν*). The obtained *b* value was determined by the slope of the log(*ν*) vs. log(*i*) graph. A *b* value of 1.0 designates a capacitive behavior and a *b* value of 0.5 indicates a diffusion-controlled behavior [41]. The obtained *b* value (Figure 4d) was 0.68. This evidence suggests that the cerium hydrogen phosphate endures a prevailing diffusion-controlled behavior for charge storage [42]. Figure 4e shows the specific capacitances of the Ce(HPO_4_)_2_.xH_2_O/NF electrode evaluated from the discharge profiles of GCD. The specific capacitance achieved a maximum as high as 114 F g^−1^ at 0.2 A g^−1^. The specific capacitance reduces steadily with increasing current density, which might be rate limited by the electrolyte ions diffusion into the electrode material. The specific capacitance retained up to 78.5 F g^−1^ at 1 A g^−1^. The obtained results indicate the promising charge storage capability and high rate ability of Ce(HPO_4_)_2_.xH_2_O.

Furthermore, to examine the real-time application of the prepared materials, Ce(HPO_4_)_2_.xH_2_O//Ce(HPO_4_)_2_.xH_2_O symmetric supercapacitor was fabricated and studied. The voltammogram plots of SSC at various sweep rates in the cut-off range between 0 and +0.8 V as shown in Figure 5a. In the same way as the three-electrode system, when increasing the sweep rate the current density also increases, and the shift of oxidation and reduction peak potential correspondingly.

Figure 5b shows the GCD profiles of the Ce(HPO_4_)_2_.xH_2_O//Ce(HPO_4_)_2_.xH_2_O symmetric device at different current densities. The shape of the CD profiles demonstrates good capacitance behavior with linear triangular profiles at the cut-off range between 0 and 0.8 V. In addition, no considerable ohmic drop was noticed in the GCD curves. Furthermore, the EIS was examined to appreciate further perceptive information of the Ce(HPO_4_)_2_.xH_2_O//Ce(HPO_4_)_2_.xH_2_O device. Figure 5c shows the EIS studies in the frequency region from 0.1 to 100 kHz at zero rest potential with the two-electrode cell set-up to distinguish the resistance or conductivity of the active material. The curve demonstrates the pseudo-capacitance behavior of the symmetric supercapacitor [43]. The inset of Figure 5c depicts the resultant equivalent circuit of the constructed SSC device. In the high-frequency range, the semicircle radius reveals the resistance of charge transfer (*R*_ct_), and ESR is attained from the intercepts of the Nyquist plot with the real part. In the low-frequency range, a straight line was obtained, which resulted in the capacitive behavior of the symmetric supercapacitor device. This performance agrees with ESR demonstrating the electrode conductivity, contact resistance, and charge transfer resistance of the electrolyte and electrode material interface. The symmetric supercapacitor established was a small radius of a semicircle and a steep straight line at high frequency. The equivalent circuit component values were *R*_s_ = 1.2 Ω, *R*_ct_ = 78.7 Ω, and *Z*_w_ = 0.22 Ω, which were lower, indicating that the 1D Ce(HPO_4_)_2_.xH_2_O with higher surface area, excellent porosity, and remarkable electrical conductivity corresponding to the rapid buffering of ion and transfer of electron in Ce(HPO_4_)_2_.xH_2_O electrode, which enhances the electrochemical performances.

Figure 5d exhibits the specific capacitances of the Ce(HPO_4_)_2_.xH_2_O//Ce(HPO_4_)_2_.xH_2_O symmetric device estimated based on the whole weight of the electrode achieves 23.44 F g^−1^ at the current density of 0.3 A g^−1^ and retains 13.95 F g^−1^ at 1 A g^−1^.

Since specific power and specific energy are dual essential aspects for estimating the electrochemical behavior of the SSC, the specific energy and specific power of the Ce(HPO_4_)_2_.xH_2_O//Ce(HPO_4_)_2_.xH_2_O symmetric supercapacitor were calculated. Figure 5e presents the Ragone plot of the fabricated Ce(HPO_4_)_2_.xH_2_O//Ce(HPO_4_)_2_.xH_2_O symmetric supercapacitor device conforming to the connection between energy density and power density. The device can distribute an energy density of 2.08 Wh kg^−1^ at a power density of 149.76 W kg^−1^ and maintain 1.24 Wh kg^−1^ at 499.88 W kg^−1^, demonstrating rapid ion-diffusion and transport of electrons and highlighting its higher power density (499.88 W kg^−1^).

Furthermore, the electrochemical stability measurement is a significant property of energy storage systems. Figure 5f shows the long-standing cyclic stability of the typical Ce(HPO_4_)_2_.xH_2_O//Ce(HPO_4_)_2_.xH_2_O symmetric supercapacitor over 5000 repeated GCD profiles at a constant current density of 1 A g^−1^. It could be seen that the capacitance still can retain 97% of its initial value after 5000 profiles, revealing its excellent stability. Even at higher current densities, the Ce(HPO_4_)_2_.xH_2_O//Ce(HPO_4_)_2_.xH_2_O symmetric supercapacitor still exhibits excellent long-standing stability. It is worth mentioning that the capacitance retention of 92.7% at 1 A g^−1^ after 5000 profiles. The outstanding cycling stability could be attributed to the good mechanical strength of 1D nanorod structure.

Figure 5g shows the EIS of Ce(HPO_4_)_2_.xH_2_O//Ce(HPO_4_)_2_.xH_2_O symmetric supercapacitor, which demonstrates the radius of the semicircle is small at high frequency and a steeper straight line at lower frequency before and after 5000 GCD cycles. In addition, over 5,000 cycles the obtained *R*_s_ value (2.14 Ω), which is slightly greater than an initial cycle of 1.73 Ω, and similarly the values of *R*_ct_ marginally improved from 72.6 Ω to 77.4 Ω. Therefore, the EIS plots too indicate that both *R*_s_ and *R*_ct_ have slight changes in the resistance during the long-term stability measurements, illustrating that the electrode material can withstand nearly unaffected ionic and electronic transport properties over a 5000 GCD cycle. The equivalent circuit element values of *R_*s*_*, *R_*ct*_*, and *Z_*w*_* are very lower, indicating that the 1D Ce(HPO_4_)_2_.xH_2_O with good specific surface area, excellent porosity, and remarkable electrical conductivity attributed to the rapid electron and ion transfer in 1D Ce(HPO_4_)_2_.xH_2_O electrode improved the electrochemical performances.

## 4. Conclusions

We successfully synthesized one-dimensional cerium hydrogen phosphate by a facile hydrothermal method for supercapacitor applications. The fabricated working electrode (Ce(HPO_4_)_2_.xH_2_O/NF) exhibited a supreme capacitance of 114 F g^−1^ at a current density of 0.2 A g^−1^ in a three-electrode system. In addition, the assembled SSC based on Ce(HPO_4_)_2_.xH_2_O//Ce(HPO_4_)_2_.xH_2_O exhibits reasonable specific energy (2.08 Wh kg^−1^), moderate specific power (499.88 W kg^−1^), and outstanding cyclic durability (retains 92.7% of its initial capacitance over 5000 GCD cycles). These results suggest that the prepared electrode material could open new avenues for energy storage applications.

## Data Availability

Not applicable.

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
