# Peer review of "Novel Supercapacitor Electrode Derived from One Dimensional Cerium Hydrogen Phosphate (1D-Ce(HPO4)2.xH2O)"

_molecules, 2022, doi:10.3390/molecules27227691_

Round 1

Reviewer 1 Report

The focus of this article is the use of Ce(HPO4) nano structures for energy storage   applications. The novelty of the article is clear enough, and manuscript is well designed. however, manuscript require improvement interms of supercapacitor performance evaluation and some critical analysis.  I do recommend this manuscript reconsider after a revision. Detail comments are included below.

1)            The superiority or critical improvements should be defined in abstract. Remove general terms

2)            Performance evaluation of the supercapacitor is over estimated because of insuitable equations. Values must be recalculate by using the integral area under the discharge curve.

Some suggested articles helpful to improve the manuscript quality.

Journal of Energy Storage - https://doi.org/10.1016/j.est.2022.104357

Journal of Power Sources - https://doi.org/10.1016/j.jpowsour.2020.228544

3)             Ce(HPO4) presented in the SEM images not uniform nanorod structure. There are particles and clustures also appear in the images.   

4)            Ce(HPO4) grown Ni-foam should be presented.

5)            TEM analysis should provide for Ce(HPO4) nanorods required.

6)            Page 10 “redox reactions are well-defined reversible” No visible redox peaks appear on the CV. Detailed description of oxidation-reduction is required.

7)            Different potential window measurements are required to understand the performance characteristics of the system.

8)            Recent articles suggest citation to improve the article overall quality.

Int. Journal of Energy Res. ; https://doi.org/10.1002/er.6916

International Journal of Hydrogen Energy ; https://doi.org/10.1016/j.ijhydene.2019.04.267

Energy Technology ; https://doi.org/10.1002/ente.201900511

Author Response

Answer to the Reviewer#1 Comments

Journal: Molecules

Manuscript ID: molecules-1963074

We thank the editors and reviewers for their valuable comments in respect of this manuscript. As per their comments and suggestions, the manuscript is thoroughly checked, appropriately modified, corrected the mistakes and new experimental results are added and discussed. The answers to the comments of the reviewers, point by point, are given as follows, and some of the answers are incorporated in the revised manuscript as well.

Reviewer#1

The focus of this article is the use of Ce(HPO4) nano structures for energy storage   applications. The novelty of the article is clear enough, and manuscript is well designed. However, manuscript requires improvement in terms of supercapacitor performance evaluation and some critical analysis.  I do recommend this manuscript reconsider after a revision. Detail comments are included below.

Comment (1): The superiority or critical improvements should be defined in abstract. Remove general terms.

Answer (1): We comply and appreciate the reviewer comment. As pee the reviewer comment, the general terms has been excluded from the abstract.

Comment (2): Performance evaluation of the supercapacitor is over estimated because of insuitable equations. Values must be re-calculated by using the integral area under the discharge curve. Some suggested articles helpful to improve the manuscript quality.

Journal of Energy Storage - https://doi.org/10.1016/j.est.2022.104357

Journal of Power Sources - https://doi.org/10.1016/j.jpowsour.2020.228544

Answer (2): We appreciate and comply with the referee’s comment. As per the reviewer suggestion, the specific capacitance has been recalculated using integral area under the discharge curve (please refer equation (1)) and also the afore-mentioned articles have been incorporated in the revised manuscript. Please see the references [19] and [20] in the revised manuscript.

Comment (3): Ce(HPO4) presented in the SEM images not uniform nanorod structure. There are particles and clusters also appear in the images.   

Answer (3): The given comment is valid and accepted. Giving due respect to reviewers comment, Ce(HPO4) exhibited nanorod structures along with particles and clusters. But the majority of the part was covered by nanorod structures. Hence, we claimed the nanorod structure. In the revised manuscript, we removed the word “uniform” and included the sentence of “Ce(HPO4) exhibited nanorod structures along with particles and clusters”.

Comment (4): Ce(HPO4) grown Ni-foam should be presented.

Answer (4): We appreciate and comply with referee’s comment. Ce(HPO4) grown nickel form has been incorporated in the revised manuscript. Please refer scheme 1 in the revised manuscript.

Comment (5): TEM analysis should provide for Ce(HPO4) nanorods required.

Answer (5): We appreciate and comply with the referee’s comment. Giving due respect to reviewer comment, in our institution we don’t have the facility of Transmission Electron Microscopy (TEM). Generally, we send samples to central instrumentation facility centre to analyse the physicochemical characterization. Normally it will take one month to analyse the particular study. Hence, we are unable to do HRTEM analysis.

Comment (6): Page 10 “redox reactions are well-defined reversible” No visible redox peaks appear on the CV. Detailed description of oxidation-reduction is required.

Answer (6): We appreciate and comply with the referee’s comment. Yes, no redox peak appeared in the cyclic voltammogram (Fig. 5(a)). We deeply apologize for the wrong statement. The following correction (statement) has been incorporated in the revised manuscript. “The shape of the CD profiles demonstrates that the good capacitance behavior with linear triangular profiles at the cut-off range between 0 and 0.8 V. In addition, no considerable ohmic drop was noticed in the GCD curves”.

Comment (7): Different potential window measurements are required to understand the performance characteristics of the system.

Answer (7): The given comment is valid and accepted. Giving due respect to reviewer comment, the potential window used in the present study for fabricated symmetric supercapacitor is only 0.0 to 0.8 V. If we operate the potential window beyond 1.0 V (for example, 0 to 1 V or 0 to 1.2 V or 0 to 1.4 V etc.), different potential window measurement is useful to evaluate the performance characteristics of the system. But, in the present system we used maximum potential of 0.8 V. Hence, we didn’t carry out the different potential window measurements.  

Comment (8): Recent articles suggest citation to improve the article overall quality.

Int. Journal of Energy Res.; https://doi.org/10.1002/er.6916

International Journal of Hydrogen Energy; https://doi.org/10.1016/j.ijhydene.2019.04.267

Energy Technology; https://doi.org/10.1002/ente.201900511

Answer (8): We thank the reviewer to identify us the above important recent articles for supercapacitor applications. We have been included these references in the revised manuscript. For your kind reference, please see the references [3], [4] and [5] in the revised manuscript.

Reviewer 2 Report

Review comment.

This paper reports the preparation of Ce(VI) hydrogen phosphate and its use in supercapacitors. The corresponding device showed well-defined behavior in electrochemical systems. But the paper is with quite a few problems that need further confirmation and clarification before it can be considered for publication in Molecules.

1.     What is the exact molecular formula for Ce(VI) hydrogen phosphate? Ce(HPO4) indicates a valence state of 2+ for Ce.

2.     Please clearly differentiate the capacitance symbol calculated from the three electrode fashion and two electrode fashion to avoid misunderstanding. Also, please clearly define the m in equation (2). Is it the weight for one electrode or for two?

3.     For EIS tests, the upper frequency limit is quite low, why not trying higher frequency like 10 MHz?

4.     The confirmation of the Ce(VI) hydrogen phosphate is still not solid. Please elaborate the XRD data a bit more, what do those peaks correspond for?

5.     Regarding the XPS, O seems to exist in only one state in the material, why it existed as two deconvolutable peaks there? Same question also goes to P2p.

6.     The authors are strongly recommended to confirm their calculation on capacitance, as well as on all the component resistances from EIS. For capacitance, normally two-electrode system usually gives a capacitance that is about 1/4 that from the three-electrode system. But the current value looks far different from that. For EIS, the charge transfer resistance is apparently larger than the given values.

7.     In page 7, the authors claimed “The well-aligned peak was noticed at 540 ?C of DTGA signifies that the Ce(HPO4) material possess high crystalline.” But from XRD, the peaks are actually not that sharp, indicating a relatively low crystallinity.

8.     In page 8, the authors used auxiliary electrode in the text. Please unify the terms in the paper.

9.     In page 10, the authors claimed that “shape of the CD profiles demonstrates almost symmetric”, which apparently is not the case from Fig. 5b.

10.  The Fig. 5d does not contain equivalent circuit.

11.  The authors showed the existence of mesopores in the synthesized material. More discussion needs to be included. The authors can check these papers for reference. Chemistry–An Asian Journal 12, no. 5 (2017): 503-506. Angewandte Chemie International Edition 47, no. 2 (2008): 373-376.

12.  There are some typos and English issues, and the authors are suggested to thoroughly confirm them. For instance, page 1, “Thus, it is (an) important to attain the unique architecture…”; Page 4, reference electrode (CE).

Author Response

Answer to the Reviewer#2 Comments

Journal: Molecules

Manuscript ID: molecules-1963074

We thank the editors and reviewers for their valuable comments in respect of this manuscript. As per their comments and suggestions, the manuscript is thoroughly checked, appropriately modified, corrected the mistakes and new experimental results are added and discussed. The answers to the comments of the reviewers, point by point, are given as follows, and some of the answers are incorporated in the revised manuscript as well.

Reviewer#2

This paper reports the preparation of Ce(VI) hydrogen phosphate and its use in supercapacitors. The corresponding device showed well-defined behavior in electrochemical systems. But the paper is with quite a few problems that need further confirmation and clarification before it can be considered for publication in Molecules.

Comment (1): What is the exact molecular formula for Ce(VI) hydrogen phosphate? Ce(HPO4) indicates a valence state of 2+ for Ce.

Answer (1): We appreciate and comply with the referee’s comment. The actual formula is Ce(HPO4)2.xH2O. The thermogram of TGA showed a weight loss around 200 °C. It confirms coordinated water.

Comment (2): Please clearly differentiate the capacitance symbol calculated from the three electrode fashion and two electrode fashions to avoid misunderstanding. Also, please clearly define the m in equation (2). Is it the weight for one electrode or for two?

Answer (2): The given comment is valid and accepted. Now we have rectified in the revised manuscript.

  • The symbol Cs stands for three-electrode system where as Ccell stands for two electrode system (symmetric supercapacitor device).
  • Now, we clearly defined the “m” in the equation (2), and
  • We have used one electrode mass to calculate the specific capacitance.

Comment (3): For EIS tests, the upper frequency limit is quite low, why not trying higher frequency like 10 MHz?

Answer (3): We comply and appreciate the referee’s comment. Giving due respect to the reviewer's comment, the electrochemical workstation is currently under repair. So, we weren't able to repeat the experiment with higher frequency. In our future experiments, we keep in mind your comment and will carry out the EIS measurement with higher frequency region. 

Comment (4): The confirmation of the Ce(VI) hydrogen phosphate is still not solid. Please elaborate the XRD data a bit more, what do those peaks correspond for?

Answer (4): Thank you very much for your valuable comment. The discussion part of XRD has been elaborated in the revised manuscript.

Comment (5): Regarding the XPS, O seems to exist in only one state in the material, why it existed as two deconvolutable peaks there? Same question also goes to P2p.

Answer (5): The given comment is valid and accepted. The two peaks of oxygen are associated with the phosphate and coordinated water. Whereas, the P 2p high resolution XPS spectrum of the Ce(HPO4) shows two deconvoluted peaks at 132.9 and 133.8 eV accountable for the P 2p3/2 and P 2p1/2 of elemental P state, and oxidized P state, respectively.

Comment (6): The authors are strongly recommended to confirm their calculation on capacitance, as well as on all the component resistances from EIS. For capacitance, normally two-electrode system usually gives a capacitance that is about 1/4 that from the three-electrode system. But the current value looks far different from that. For EIS, the charge transfer resistance is apparently larger than the given values.

Answer (6): The given comment is valid and accepted. As per the reviewer's suggestion, the specific capacitance (Cs) calculation has been recalculated using integral area under the discharge curve (three electrode system; Fig. 4(b)) and the values are depicted in Fig. 4(e). Now, the two-electrode system showing the specific capacitance (Ccell) is almost ¼ of the three-electrode system. 

Yes, you are correct. The charge-transfer resistance is re-calculated and incorporated in the revised manuscript.  

Comment (7): In page 7, the authors claimed “The well-aligned peak was noticed at 540 deg. C of DTGA signifies that the Ce(HPO4) material possess high crystalline.” But from XRD, the peaks are actually not that sharp, indicating a relatively low crystallinity.

Answer (7): We comply and appreciate the referee’s comment. Giving due respect to reviewer comment, the word “high” has been removed in the revised manuscript.

Comment (8): In page 8, the authors used auxiliary electrode in the text. Please unify the terms in the paper.

Answer (8): The given comment is valid and accepted. Now we have rectified as “counter electrode (CE)” in the revised manuscript.

Comment (9): In page 10, the authors claimed that “shape of the CD profiles demonstrates almost symmetric”, which apparently is not the case from Fig. 5b.

Answer (9): We appreciate and comply with the referee’s comment. Yes, no redox peak appeared in the cyclic voltammogram (Fig. 5(a)). We deeply apologize for the wrong statement. The following correction (statement) has been incorporated in the revised manuscript. “The shape of the CD profiles demonstrates that the good capacitance behavior with linear triangular profiles at the cut-off range between 0 and 0.8 V. In addition, no considerable ohmic drop was noticed in the GCD curves”.

Comment (10): The Fig. 5d does not contain equivalent circuit.

Answer (10): We comply and appreciate the referee’s comment. Giving due respect to reviewer comment, we wrongly mentioned it. It was 5(c) not 5(d). As per the reviewer suggestion, the equivalent circuit has been included in the Fig. 5(c).

Comment (11): The authors showed the existence of mesopores in the synthesized material. More discussion needs to be included. The authors can check these papers for reference. Chemistry–An Asian Journal 12, no. 5 (2017): 503-506. Angewandte Chemie International Edition 47, no. 2 (2008): 373-376.

Answer (11): The given comment is valid and accepted. As per the reviewer suggestion, the discussion part of nitrogen sorption isotherm has been improved and also the above mentioned articles have been cited in the revised manuscript. Please see references [29], and [30].

Comment (12): There are some typos and English issues, and the authors are suggested to thoroughly confirm them. For instance, page 1, “Thus, it is (an) important to attain the unique architecture…” Page 4, reference electrode (CE).

Answer (12): The given comment is valid and accept. We sincerely apologise for these mistakes. The manuscript was thoroughly checked for grammatical error/mistakes/typo errors and it has been rectified in the revised manuscript.

Round 2

Reviewer 1 Report

All comments are successfully addressed by the authors. Accept

Author Response

Answer to the Reviewer#1 Comments

 Journal: Molecules

Manuscript ID: molecules-1963074

Reviewer#1

Comment (1): All comments are successfully addressed by the authors. Accept

Answer (1): We are very much thankful to the reviewer accepting our research article in esteemed molecules journal.

Reviewer 2 Report

The authors have made corresponding improvement as suggested. Here some comments based on the authors’ response.

1. Page 10, in the caption of Figure 5, equivalent circuit is in (c), not (g)

2. Please carefully check the format and arrangement of the references. Currently, there are clear mistakes there with the alignment.

3. In page 6, citing is necessary in clarifying the XRD peaks.

4. Why do authors use “Ce(HPO4)” in the paper? This is so much misleading in terms of stoichiometric balance of elements.

5. The equation for calculating capacitance is not correct. For three electrode C= I ?? /???, here m is the weight for the single electrode, where as for 2 electrodes system, the equation is essential same as C= I ?? /???, where m should be the weight for two electrodes summed. I recommend the authors to double confirm their calculations.

Author Response

Answer to the Reviewer#2 Comments

Journal: Molecules

Manuscript ID: molecules-1963074

We thank the editors and reviewers for their valuable comments in respect of this manuscript. As per their comments and suggestions, the manuscript is thoroughly checked, appropriately modified, corrected the mistakes and new experimental results are added and discussed. The answers to the comments of the reviewers, point by point, are given as follows, and some of the answers are incorporated in the revised manuscript as well.

Reviewer#2

The authors have made corresponding improvement as suggested. Here some comments based on the authors’ response.

Comment (1): Page 10, in the caption of Figure 5, equivalent circuit is in (c), not (g).

Answer (1): The given comment is valid and accepted. We sincerely apologise for this mistake. It was typo error and it has been rectified in the revised manuscript.

Comment (2): Please carefully check the format and arrangement of the references. Currently, there are clear mistakes there with the alignment.

Answer (2): We comply and appreciate the referee’s comment. The reference section has been re-arranged as per the journal format.

Comment (3): In page 6, citing is necessary in clarifying the XRD peaks.

Answer (3): The given comment is valid and accepted. The citations have been incorporated in the revised manuscript. Please refer [26, 27] references in the revised manuscript.

Comment (4): Why do authors use “Ce(HPO4)” in the paper? This is so much misleading in terms of stoichiometric balance of elements.

Answer (4): We comply and appreciate the referee’s comment. The correct formulae (stoichiometric balanced), Ce(HPO4)2.xH2O has been added throughout the revised manuscript.

Comment (5): The equation for calculating capacitance is not correct. For three electrode C= I ?? /???, here m is the weight for the single electrode, whereas for 2 electrodes system, the equation is essential same as C= I ?? /???, where m should be the weight for two electrodes summed. I recommend the authors to double confirm their calculations.

Answer (5): We appreciate and comply with the referee’s comment. The device capacitance (symmetric supercapacitor) is re-calculated with the formula of Ccell= I ?? /??? (without multiplying factor of 2), considering “m” as the weight of two electrodes and it was depicted in the revised manuscript.

Giving due respect to reviewer comment, for 3-electrode system we used the formulae of to re-calculate the specific capacitance (as suggested by reviewer #1) values by using the integral area under the discharge curve. Since, the GCD plots were not linear.
